# Emergent flat band and topological Kondo semimetal driven by orbital-selective correlations

Lei Chen[1], Fang Xie[1], Shouvik Sur [1], Haoyu Hu[1,2], Silke Paschen [1,3], Jennifer Cano [4,5] & Qimiao Si [1] ✉

Flat electronic bands are expected to show proportionally enhanced electron correlations, which may generate a plethora of novel quantum phases and unusual low-energy excitations. They are increasingly being pursued in $d$-electron-based systems with crystalline lattices that feature destructive electronic interference, where they are often topological. Such flat bands, though, are generically located far away from the Fermi energy, which limits their capacity to partake in the low-energy physics. Here we show that electron correlations produce emergent flat bands that are pinned to the Fermi energy. We demonstrate this effect within a Hubbard model, in the regime described by Wannier orbitals where an effective Kondo description arises through orbital-selective Mott correlations. Moreover, the correlation effect cooperates with symmetry constraints to produce a topological Kondo semimetal. Our results motivate a novel design principle for Weyl Kondo semimetals in a new setting, viz. $d$-electron-based materials on suitable crystal lattices, and uncover interconnections among seemingly disparate systems that may inspire fresh understandings and realizations of correlated topological effects in quantum materials and beyond.

Certain crystalline lattices feature flat bands, via frustration caused by destructive interference in electron motion[1], which are increasingly being explored in $d$-electron-based systems[2,3]. The reduced bandwidth correspondingly enhances the effect of electron correlations. In addition, such flat bands are often topologically nontrivial. As such, these systems represent a new platform to uncover novel physics for both correlation and topology as well as their interplay[4]. For example, kagome metals may host flat bands and have been the subject of considerable recent interest for realizing unusual forms of charge-density-wave order[5–10]. They have also been implicated to exhibit a type of strange metal behavior[11–14] that resembles what has been extensively studied in quantum-critical heavy fermion metals[4,15,16].

In order to strongly influence the low-energy physics, the flat bands need to be placed near the Fermi energy. However, this typically is not the case at the level of bare (noninteracting) electron band structure. There have been considerable recent experimental efforts to tune the bare flat bands to the vicinity of the Fermi energy. With the rare exception coming from materials search[13], the tuning study has met with only limited success[17,18]. Because the flat bands are associated with $d$-electrons, the energy scales that determine the flat band placement (with respect to the Fermi energy) are relatively large, and, as a result, it is challenging to achieve the required tuning. We are thus motivated to ask the following important questions: Can electron correlations generate emergent flat bands at the Fermi energy in $d$-

[1]Department of Physics and Astronomy, Rice Center for Quantum Materials, Rice University, Houston, TX 77005, USA. [2]Donostia International Physics Center, P. Manuel de Lardizabal 4, 20018 Donostia-San Sebastian, Spain. [3]Institute of Solid State Physics, Vienna University of Technology, Wiedner Hauptstr. 8-10, 1040 Vienna, Austria. [4]Department of Physics and Astronomy, Stony Brook University, Stony Brook, NY 11794, USA. [5]Center for Computational Quantum Physics, Flatiron Institute, New York, NY 10010, USA. ✉e-mail: qmsi@rice.edu

electron-based systems? And, if so, to what extent do the resulting phases display nontrivial correlation and/or topological physics?

We address both issues in a Hubbard model in which the non-interacting limit features a topologically nontrivial flat band that is far away from the Fermi energy. Due to a well-separated hierarchy in the widths of the flat band and wide bands that it is coupled to, and through the formation of compact molecular orbitals[19], orbital-selective Mott correlations develop[11]. We show that such orbital-selective correlations lead to emergent flat bands that are pinned to the Fermi energy. Moreover, using symmetry constraints in interacting settings[20], which are based on Green's function eigenstates (as opposed to Bloch states[21–25]), we demonstrate that the emergent flat bands lead to a topological Kondo semimetal. The latter is in the same family as Weyl Kondo semimetals that appear in topological Kondo-lattice models[26,27] and materials[28,29] of both existing and designed[20,30] heavy fermion systems. The qualitative physics is illustrated in Fig. 1a–c. Importantly, our approach is based on an exact construction of molecular orbitals and an exact mapping to a heavy fermion description; this is in addition to the exact constraints that symmetry places, which will also come into our analysis. Our results motivate a design principle for the Weyl Kondo semimetals in the new setting of $d$-electron-based systems and point to the realization of fractional Chern insulators[31] in transition-metal compounds.

## Results

### One-orbital Hubbard model on the clover lattice

For a proof-of-principle demonstration, we consider a variant of the kagome lattice, the two-dimensional (2D) clover lattice. As shown in Fig. 1d, it contains five sublattices per unit cell. Leaving the details of the model to be given in the Methods and in Supplementary Note 1, we note that this lattice features a flat band (Supplementary Note 2). As a case study, the model is simplified while preserving the topological nature of the flat band; we do so by removing the $C_3$ symmetry of the clover lattice (Methods), leaving only a mirror symmetry $M_x$. There is one orbital per site. The Hubbard model takes the form $\mathcal{H} = \mathcal{H}_0 + \mathcal{H}_1$, where $\mathcal{H}_0$ is the kinetic term and $\mathcal{H}_1$ represents the onsite Hubbard interaction. We consider the generic setting that has not been analyzed before, namely with the flat band of the noninteracting Hamiltonian being far away from the Fermi energy, as illustrated in Fig. 1a and shown in Fig. 2a.

The lattice can be divided into two groups of sublattices (denoted by blue and yellow dots in Fig. 1d), which contain different numbers of sites per unit cell. The flat-band formation can be seen by considering only the nearest neighbor hopping between the blue and yellow sites, reflecting a destructive interference of the electronic wavefunction on the lattice[32] (see Supplementary Note 2). The flat band overlaps with the wide bands.

### Molecular orbitals, effective extended Hubbard model, and the solution method

A flat band that is topologically nontrivial cannot by itself be represented by exponentially localized symmetry-preserving (Kramers-doublet) Wannier orbitals[25]. Such a Wannierization only becomes possible when other bands are considered along with the flat band. In addition, a flat band coming from destructive interference comprises

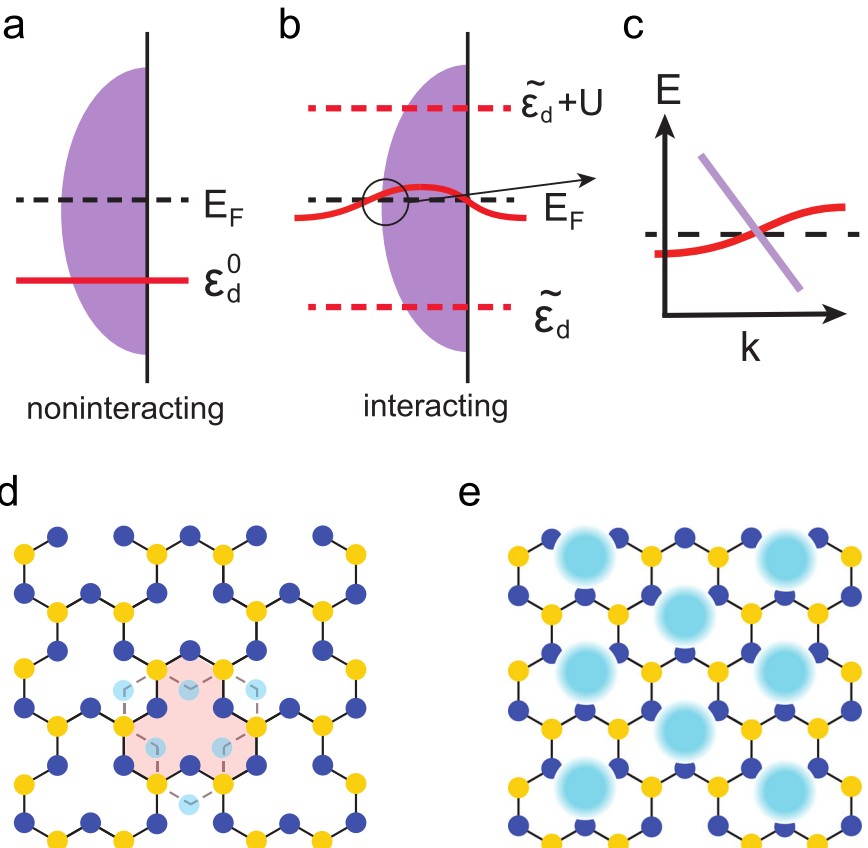

**Fig. 1 | Illustration of the bare and emergent flat bands and lattice geometry. a** In the noninteracting case, a flat band (red solid line) appears far away from the Fermi energy. **b** In the presence of orbital-selective correlations, an interaction-driven flat band emerges at the Fermi energy (red solid line), while leaving incoherent excitations far away from the Fermi energy (red dashed lines). **c** The emergent flat band crosses a dispersive band, leading to a topological Kondo semimetal with symmetry-protected Dirac/Weyl nodes that are pinned close to the Fermi energy, within an effective Kondo energy scale. **d** Geometry of the clover lattice with 5 sublattices per unit cell. The lattice does not have inversion symmetry. This can be seen from the mismatch between the (dark) blue sublattices and their inversion counterparts (dots in light blue). **e** The Wannier orbitals are near the geometric centers (shaded blue circles) of the unit cells, which form a triangular lattice.

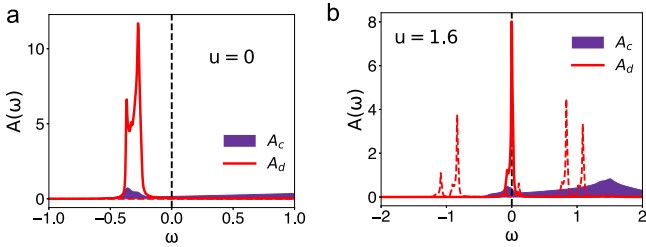

**Fig. 2 | Bare and emergent flat bands. a** The DOS of the $d$ and $c$ electrons for the noninteracting case. **b** The corresponding result at $u = 1.6$; here the red solid (dashed) lines denote the coherent (incoherent) part of the $d$-electron excitations. The purple backgrounds mark the DOS of the conduction $c$ electrons.

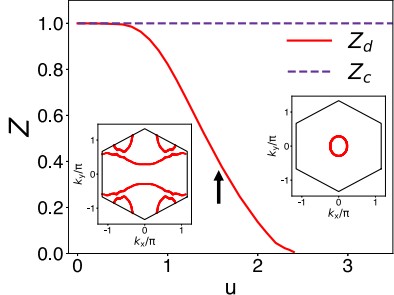

**Fig. 3 | Orbital-selective correlations.** The quasiparticle weights of the $d$ and $c$ electrons, $Z_d$ and $Z_c$, as a function of the effective interaction $u$. The left (right) inset plots the Fermi surface for the values of interactions right below (above) $u_c = 2.4$.

states from multiple (inequivalent) atomic sites. If one Wannier orbital is to primarily capture this flat band, this Wannier orbital (and, by extension, the others accompanying it) must involve multiple atomic orbitals. In other words, in this case, the Wannier orbitals are necessarily molecular orbitals. We again stress that the mapping we use is exact.

In our case, we can restrict to three bands (see Supplementary Note 3). We find the centers of the three localized Kramers-doublet Wannier orbitals to be located near the geometric center of the unit cell, which forms a triangular lattice (see Fig. 1e). Importantly, one Wannier orbital primarily captures the flat band[11]; it is the most localized and is denoted as the $d$ orbital. The other two Wannier orbitals are dominated by the wide bands; they decay much more slowly and are marked as $c$ orbitals. The large difference in the width of the flat band ($D_{\text{flat}}$) and the wide bands ($D_{\text{wide}}$) opens up a range of interactions that are in between. In this range, the electron correlations are strongly orbital-selective and the system affords a Kondo/Anderson-lattice model description[11].

We project the Hubbard model of the original lattice to the Wannier basis. This leads to the effective model expressed in terms of the $d$ and $c$ Wannier orbitals with $H_{\text{eff}} = H_0 + H_{\text{int}}$. The kinetic term is specified in the Methods. For the interaction terms, we keep the most dominant interactions on the Wannier basis. They include the onsite Hubbard interaction among the $d$ electrons and the density-density interactions between the $d$ and $c$ electrons:

$$H_{\text{int}} = H_d + H_F$$
$$= \sum_i \frac{u}{2}\left(n_{i\uparrow}^d + n_{i\downarrow}^d - 1\right)^2 + \sum_{i,\alpha} F_\alpha\, n_i^d\, n_i^{c_\alpha} \tag{1}$$

where $n_{i,\sigma}^a = a_{i\sigma}^\dagger a_{i\sigma}$, with $a = d, c_\alpha$, $\alpha = 1, 2$, and $n_i^a = \sum_\sigma n_\sigma^a$. The onsite Hubbard interaction on the $d$-orbital, $u$, is the most dominant one, given the much more localized nature of this Wanner orbital. The density-density interactions between the $d$ and $c$ electrons, $F_\alpha$, are weaker but also sizable: $F_1/u \approx 0.3$ and $F_2/u \approx 0.25$. These effective interaction parameters are determined by those of the original Hubbard model (see Supplementary Note 3). The interactions among the $c$ electrons are relatively small compared to their bandwidths and, accordingly, will be unimportant.

To take into account the effect of the interactions, we use the U(1) slave spin (SS) method[33]. Given that only the onsite interaction of the $d$ orbital is important, we need to introduce an SS representation for the $d$ orbital only: $d_{i\sigma}^\dagger = o_{i\sigma}^\dagger f_{i\sigma}^\dagger$, where the auxiliary bosonic and fermionic operators, $o^\dagger$ and $f_\sigma^\dagger$, carry the charge and spin degrees of freedom, respectively. We treat the SS formulation at the saddle-point level and self-consistently solve the corresponding Hamiltonians for the SS and the auxiliary fermion parts. The SS method is also used to obtain the contributions to the single-electron excitations from the (interaction-driven) incoherent part of the spectrum. The details are found in the Methods and Supplementary Note 6.

## Emergent flat band at the Fermi energy

We are now in position to discuss the effect of interactions on the single-electron excitations. Consider first the density of states (DOS). In the noninteracting case, as shown in Fig. 2a, the $d$ electron DOS (the red curve) has a sharp peak compared with the background (purple color) $c$ electron component. This reflects the $d$ electrons as primarily describing the flat band. As can be seen, the peak is located far away from the Fermi level, which also reflects its origin from the noninteracting flat band (see Supplementary Note 1).

Importantly, under the influence of electron correlations, a new flat band emerges. This is demonstrated in Fig. 2b with $u = 1.6$. The emergent flat band is pinned to the Fermi energy, as captured by the coherent peak (the red solid lines) in the DOS. The background DOS associated with the conduction $c$ electrons is largely unchanged from its noninteracting counterpart. Varying the interaction strongly influences the spectral weight of the emergent flat band (see Supplementary Note 7): This part of the spectral weight is reduced as the interaction increases (comparing Fig. 2b and Supplementary Fig. S5a); the reduced spectral weight is transferred to the incoherent part (the red dashed lines). This form persists until the weight of the coherent peak is completely lost and the system goes through an orbital-selective Mott transition. When that happens, the incoherent parts of the single-electron excitations develop into the full-fledged lower and upper Hubbard bands (see Supplementary Note 7 and Fig. S5b).

## Orbital-selective Mott correlations

To expound the origin of the emergent flat band, we further analyze the orbital-selective Mott correlations in the regime of interactions of our interest, viz. $D_{\text{flat}} < u < D_{\text{wide}}$. As shown in Fig. 3, the metal-to-insulator transition of the $d$ electrons occurs at $u_c = 2.4$. We reiterate that the range of the interactions being considered here is weaker than the width of the wide bands associated with the $c$ electrons (see Supplementary Fig. S3). Thus, the $c$ electrons are fully itinerant. This justifies the neglect of the interactions among the $c$ electrons, as we have done, so that the quasiparticle weight of the $c$-electrons remains to be 1 as seen in Fig. 3. The existence of an orbital-selective Mott transition is further illustrated in the nature of the Fermi surface. As shown in Fig. 3 (insets), the Fermi surface undergoes a dramatic change across the transition. This change parallels the electron localization-delocalization (Kondo destruction) physics of heavy fermion systems[34–38]. The phase with the $d$-electrons being itinerant corresponds to the Kondo-screened phase, in which the local moment is converted into (fragile, or heavy) electronic excitations that hybridize with the conduction electrons to form the quasiparticles. By contrast, the orbital-selective Mott phase (OSMP) is analogous to the Kondo-destroyed phase of the heavy fermion systems, in which the Fermi surface is formed entirely from the conduction electrons. The fact that our noninteracting flat band is, to begin with, far away from the Fermi energy makes the parallel with the heavy Fermion systems especially

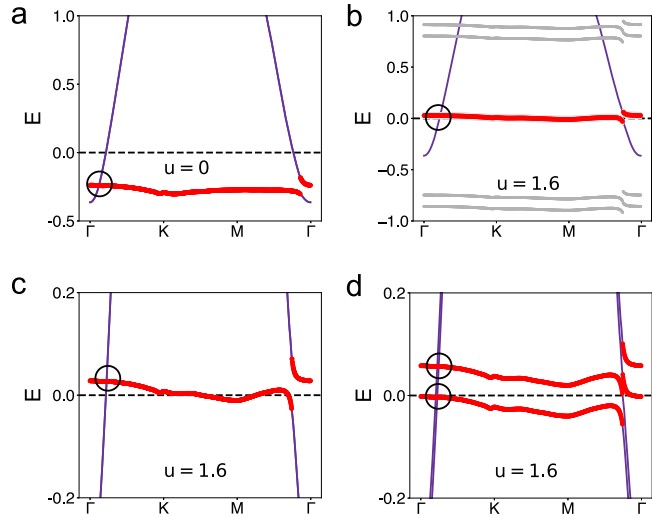

**Fig. 4 | Topological Kondo semimetal. a** The noninteracting band structure. **b** The dispersion of the single-electron excitations at $u = 1.6$. The red solid curve denotes the emergent flat band close to the Fermi energy. The grey lines mark the incoherent single-electron excitations. **c** The zoomed-in view of the emergent flat band. **d** The band structure at $u = 1.6$ with a Zeeman splitting $m_z = 0.03$.

clear. Our analysis of this flat-band system provides a realization of the Kondo physics in a one-band Hubbard model that physically describes a $d$-electron-based system.

More specifically, the value of the interaction illustrated in Fig. 2b is marked by an arrow in Fig. 3. The differentiation between the quasiparticle weights of the $d$ and $c$ electrons at this interaction characterizes the orbital-selective nature of the electron correlations. This is reflected in the $d$-electron spectral weight: as seen in Fig. 2b, the incoherent peaks (the red dashed lines) are well formed, which corresponds to the precursor of the lower and upper Hubbard bands of the OSMP (see Supplementary Fig. S5b). The coherent spectral weight, i.e., the central peak (the red solid line of Fig. 2b) is thus described in terms of the Kondo resonance of a Kondo-lattice model, in which the local moments correspond to the effective spin degrees of freedom associated with the lower and upper Hubbard bands. This description makes precise the notion that the flat band at the Fermi energy is emergent, driven by the orbital-selective Mott correlations.

## Topological Kondo semimetal

The energy dispersion of the electronic states is shown in Fig. 4. From the dispersion of the interacting ($u = 1.6$) case, we again see that a Fermi-energy-bound flat band emerges in the interacting case.

We are then in a position to analyze the symmetry constraints[21–25]. In the noninteracting limit, the three Wannier orbitals have different $M_x$ eigenvalues. The flat $d$ orbital has the $M_x$ eigenvalue +1, while the two $c$ orbitals have the $M_x$ eigenvalues of −1 and +1 respectively[11]. In the presence of time-reversal symmetry, along the $\Gamma - K$ line, the flat band from the $d$ orbital has a symmetry-protected Dirac crossing with the $c$ orbital of the opposite mirror eigenvalue. This same symmetry constraint also applies to the Kondo-driven flat band. The Dirac node for the emergent flat band is shown in Fig. 4b. A node from the flat band close to the Fermi energy allows a high tunability. As shown in Fig. 4d, a relatively small Zeeman coupling (illustrated here with $m = 0.03$, which is small compared to the width of the emergent flat band of ~0.1) causes a substantial separation of the nodes. These nodes now have two-fold degeneracy.

The orbital-selective Mott correlations are caused by local correlations. While we have provided a case study of how such correlations give rise to emergent flat bands in a particular 2D model, a similar conclusion is expected in general cases, including for models in three

dimensions (3D). There is an important distinction though. In 3D, topological nodes develop in the presence of SOC under symmetry constraints[25,39]. For noncentrosymmetric systems, or for centrosymmetric systems with the breaking of time-reversal symmetry, we can then expect the emergent flat bands to feature Weyl nodes leading to a Weyl Kondo semimetal.

To further expound on the generality of our theoretical results, we note that the metallic regime with strong orbital-selective correlations can be viewed through the Kondo analogy. From this perspective, the emergent flat band describes low-energy coherent electronic excitations associated with the Kondo-driven composite fermions. Because low-energy electronic excitations are always Fermi-energy bound, and also based on the well-established understanding that Kondo-driven composite fermions occur in the immediate vicinity of the Fermi energy, the emergent flat bands that develop through our proposed mechanism must be pinned near the Fermi energy. This represents a general principle. To explicate on this generality, we have mapped out a phase diagram to show that the proposed mechanism operates over an extended region in the $u$-$\epsilon_d^0$ parameter space (region "II" of the phase diagram given in Supplementary Note 8 and Fig. S6). Furthermore, we have carried out related calculations in a more general setting and found a similar development of an emergent flat band when the noninteracting flat band is located substantially away from the Fermi energy; the details of this analysis appear in Supplementary Note 9 and Fig. S7.

## Design principle for Weyl Kondo semimetals in physical $d$-electron systems

Weyl Kondo semimetals have so far been explored in $f$-electron-based materials, $Ce_3Bi_4Pd_3$[28,29], and several newly proposed Ce-, Pr- and U-based compounds[20,30]. The present work leads us to propose a design principle for realizing Weyl Kondo semimetals in a new setting. Importantly, our theoretical results are expected to be robust against the effect of the residual interactions among the quasiparticles. This is so because the origin of the topological nodes lies in the symmetry constraints, which have recently been shown to operate on the eigenvectors of the matrix associated with the *exact* single-electron Green's function of an interacting system[20].

Accordingly, our proof-of-principle demonstration enables us to advance a new materials design procedure for Weyl Kondo semimetals in the setting of $d$-electron-based systems. The procedure would start from 3D lattices that can host flat bands from quantum interference. Examples include the pyrochlore lattice[40], the perovskite lattices[41], and other 3D versions of the bipartite crystalline lattices[42]. We seek materials with $d$-elements and utilize orbital-selective correlations to drive interacting flat bands that are Fermi-energy-bound. Symmetry constraints can then lead to either Dirac or (with the breaking of inversion or time-reversal symmetry) Weyl nodes in these emergent flat bands. The latter case corresponds to a Weyl Kondo semimetal.

The procedure for this materials identification approach goes beyond that for Weyl Kondo semimetals in $f$-electron-based systems[30]. In addition to the requirement for both correlations and crystalline symmetry constraints, it also involves the crystal lattice conditions for the formation of flat bands in the bare dispersion. We reiterate that the noninteracting flat bands are not required to be near the Fermi energy. This is an important feature in the proposed materials design principle, given that the noninteracting flat bands in relevant materials are generically away from their Fermi energy.

## Implications for fractional Chern insulators

Fractional Chern insulators, with a fractional quantum Hall effect and the associated fractional charge in a lattice setting, have been proposed in correlated models with an appropriate (such as 1/3) filling ratio of a flat band when the latter crosses the Fermi energy[43–45]. Experimental evidence has recently been identified in twisted bilayer graphene[31], in which the moiré bands are located near the Fermi energy, in a small

external magnetic field. Our results on the Fermi-energy-bound emergent flat bands raise the possibility of another potential platform to realize the fractional Chern insulators, namely in $d$-electron-based 2D systems. Indeed, when a spin-orbit coupling is included in the 2D model, the Dirac node is gapped leading to flat $Z_2$ topological bands (see Supplementary Note 4). The residual interactions (which develop beyond the saddle-point analysis in the slave-spin approach that we have carried out) could be ferromagnetic (see Supplementary Note 5). In that case, the flat band can develop a nonzero Chern number and can be analyzed for a lattice realization of fractional quantum Hall effect[46]. Indeed, the combination of the flatness of the associated bands (see Supplementary Fig. S4) and the aforementioned residual interactions among the heavy quasiparticles represents a condition that is similar to what happens in the moiré systems[31]; however, the $Z_2$ nature of the flat bands makes them distinct and rare[47]. Accordingly, with appropriate fillings, our results suggest that the corresponding $d$-electron-based 2D materials provide a new setting for realizing a fractional Chern insulator. The naturalness of the emergent flat band crossing the Fermi energy makes our proposal robust. Thus, this represents a promising new direction for a systematic examination.

## Discussion

Our work opens a new bridge between topological flat bands and correlation physics. The interaction effect tends to localize the molecular orbital that has the most overlap with the flat band. As a result, these molecular orbitals play the role of local moments, by analogy with the local spins of Kondo systems. Our work provides a rare non-perturbative way to address the interplay between correlations and topology effects in such flat band systems and a variety of correlated materials[48]. As such, it promises to elucidate the physics of correlated kagome transition-metal compounds[2,3,13,14] and related materials. We also expect that our analysis will inspire new understandings of the correlation effects in moiré structures, which are increasingly being viewed from a Kondo perspective[49–53], as well as in other flat band systems[54]. Our work has also allowed us to advance a new materials design principle to identify Weyl Kondo semimetals in the new setting of $d$-electron-based systems. We expect the interconnections that our work reveals among seemingly disparate systems to inspire new realizations and understandings of correlated topological effects in a wide variety of quantum materials and structures. Finally, we note that our theoretical result for the emergent flat band is now supported by experiment: In a frustrated-lattice material, an emergent flat band has been observed by angle-resolved photoemission spectroscopy at the Fermi energy, even though the ab initio noninteracting band structure predicts a flat band that is considerably away from the Fermi energy[12].

## Methods

### Hubbard model on the clover lattice

The clover lattice has been discovered in real materials such as the van der Waals system $Fe_5GeTe_2$[32,55]. As shown in Fig. 1d, it contains five sites in each unit cell, which are reclassified into two groups as marked by the yellow and blue colors. For an illustrative purpose, we restrict our model to have only a $d_{z^2}$ orbital on each site. The case with other $d$-orbitals, such as $d_{xz}/d_{yz}$, have a similar realization of the geometry-induced flat bands[32]. We consider the Hubbard model written as $\mathcal{H} = \mathcal{H}_0 + \mathcal{H}_1$, where $\mathcal{H}_0$ is the kinetic term that connects the two different groups of sublattices and $\mathcal{H}_1$ represents the onsite one-orbital Hubbard interaction. We label the orbitals based on the group of sublattices to be $A/B$ and $C/D/E$ respectively. For each site, we consider the onsite Hubbard interaction,

$$\mathcal{H}_1 = U \sum_{i,\alpha} n^{\eta_\alpha}_{i,\uparrow} n^{\eta_\alpha}_{i,\downarrow},$$

(2)

where $\eta_\alpha$ ($\alpha = 1 \sim 5$) goes through all the five orbitals in each unit cell. The kinetic Hamiltonian is written as

$$\mathcal{H}_0 = \sum_{ij,\alpha\beta,\sigma} t\eta^\dagger_{i\alpha\sigma}\eta_{j\beta\sigma} - \mu_0 \sum_{i\alpha\sigma} \eta^\dagger_{i\alpha\sigma}\eta_{i\alpha\sigma} + \sum_{i,\sigma,\alpha\in\{C,D,E\}} m\eta^\dagger_{i\alpha\sigma}\eta_{i\alpha\sigma} \\ + \sum_{i\sigma\alpha\in\{D,E\}} \gamma\eta^\dagger_{i\alpha\sigma}\eta_{i\alpha\sigma}.$$

(3)

Here $t$ denotes the nearest neighbor hopping between the two sites that are connected by the solid lines shown in Fig. 1d, and $\mu_0$ is the chemical potential. In addition, $m$ denotes the energy splitting between the two groups of sublattices, which have a different local environment and thus generically have different energy levels. Finally, $\gamma$ represents an additional energy splitting between $C$ and $D/E$, which breaks the $C_3$ rotational symmetry. As mentioned earlier, we work with the case that breaks $C_3$ symmetry to simplify the symmetry characterization and, thus, the Wannier construction. It is possible that this $C_3$ symmetry breaking spontaneously appears as a result of interactions that drive a nematic order, although, for our illustrative purpose, we do not pursue this route specifically. A detailed analysis of the dispersion is shown in Supplementary Note 1.

### Effective extended Hubbard model

We project the Hubbard model of the original lattice to the Wannier basis. This leads to the effective model expressed in terms of the $d$ and $c$ Wannier orbitals with $H_{eff} = H_0 + H_{int}$. The kinetic term takes the following form:

$$H_0 = H_d + H_c + H_V \\ = \sum_{ij,\sigma} t_{ij}\left(d^\dagger_{i\sigma}d_{j\sigma} + h.c.\right) - \sum_i \mu d^\dagger_{i\sigma}d_{i\sigma} \\ + \sum_{ij,\alpha\beta,\sigma} t^{\alpha\beta}_{ij}\left(c^\dagger_{i\alpha\sigma}c_{j\beta\sigma} + h.c.\right) + \sum_{i\alpha\sigma}(\Delta_\alpha - \mu)c^\dagger_{i\alpha\sigma}c_{i\alpha\sigma} \\ + \sum_{ij,\alpha\sigma} V^\alpha_{ij}\left(d^\dagger_{i\sigma}c_{j\alpha\sigma} + h.c.\right).$$

(4)

Here, $d^\dagger_{i\sigma}$ ($c^\dagger_{i\alpha\sigma}$) creates a heavy (light) electron at the position $i$ with spin $\sigma$ (and orbital $\alpha$). In addition, $t_{ij}$ ($t^{\alpha\beta}_{ij}$) denotes the hopping parameter between the $d$ ($c$) electrons at the positions $i$ and $j$ (orbitals $\alpha$ and $\beta$). Moreover, $V^\alpha$ represents the hybridization between the light orbital $\alpha$ and the heavy orbital. Finally, $\Delta_\alpha$ describes the difference in the energy levels between the $d$ orbital and $c$ orbitals, and $\mu$ specifies the chemical potential. We will focus on the case with the $d$-electron level being deep below the Fermi energy and will show that the interaction effect creates a heavy band near the Fermi energy. The noninteracting dispersion is shown in Fig. 4a, where there is a Dirac crossing between the flat band and lower wide band located deep below the Fermi energy; the crossing is protected by the $M_x$ lattice symmetry[25].

### Slave spin method and self-consistent equations

We describe the U(1) slave spin approach[33]. Because the bandwidth of the heavy orbital is much smaller than those of the light orbitals, the interaction effect is most pronounced on the $d$ orbital. We therefore only introduce the SS representation on the $d$ orbital: $d^\dagger_{i\sigma} = o^\dagger_{i\sigma}f^\dagger_{i\sigma}$. The auxiliary bosonic field $o^\dagger_\sigma = P^+ S^+_\sigma P^-$ is represented by the spin operator accompanied with the projection operators $P^\pm_{i\sigma} = \frac{1}{\sqrt{1/2 \pm S^z_{i\sigma}}}$ suitable for a system that is away from half filling. We treat the SS formulation at the saddle-point level by fully decoupling the SS and auxiliary fermion

operators. This leads to the decoupled Hamiltonian:

$$H^f = \sum_{\mathbf{k},\sigma} \langle O_\sigma \rangle^2 \epsilon_{\mathbf{k}}^f f_{\mathbf{k}\sigma}^\dagger f_{\mathbf{k}\sigma} + \sum_{\mathbf{k}\sigma} \left( -\mu - \lambda_\sigma + \lambda_\sigma^0 \right) f_{\mathbf{k}\sigma}^\dagger f_{\mathbf{k}\sigma}$$
$$+ \sum_{\mathbf{k},\alpha\sigma} \left( V_{\mathbf{k}}^\alpha \langle O_\sigma^\dagger \rangle f_{\mathbf{k}\sigma}^\dagger c_{\mathbf{k}\alpha\sigma} + h.c. \right) + H_c$$
$$H^S = \sum_\sigma \left[ \tilde{\epsilon}_f \left( \langle O_\sigma \rangle O_\sigma^\dagger + h.c. \right) + \sum_\alpha \left( \tilde{V}_\alpha O_\sigma^\dagger + h.c. \right) \right] \quad (5)$$
$$+ \sum_\sigma \lambda_\sigma \left( S_\sigma^z + \frac{1}{2} \right) + H_{\text{int}}^S \,,$$

where

$$H_{\text{int}}^S = \frac{u}{2} \left( \sum_\sigma S_\sigma^z \right)^2 + \sum_\alpha F_\alpha \langle n_c^\alpha \rangle \sum_\sigma S_\sigma^z, \quad (6)$$

and $\epsilon_f(\mathbf{k})$ and $V_{\mathbf{k}}^\alpha$ are the Fourier transforms of $t_{ij}$ and $V_{ij}^\alpha$, respectively, $\tilde{\epsilon}_f = \sum_{\mathbf{k}} \epsilon_f(\mathbf{k}) \langle f_{\mathbf{k}}^\dagger f_{\mathbf{k}} \rangle$, and $\tilde{V}_\alpha = \sum_{\mathbf{k}} V_{\mathbf{k}}^\alpha \langle f_{\mathbf{k}}^\dagger c_{\mathbf{k}} \rangle$. In addition, $O_\sigma = \langle P^+ \rangle S_\sigma^+ \langle P^- \rangle$, $\lambda_\sigma^0 = 2 \left( \bar{\epsilon}_f + \sum_\alpha \bar{V}_\alpha \right) \eta_\sigma$ with $\bar{\epsilon}_f = \tilde{\epsilon}_f \langle O_\sigma \rangle \langle O_\sigma^\dagger \rangle + c.c.$, $\bar{V} = \tilde{V} \langle O_\sigma^\dagger \rangle + c.c.$ and $\eta_\sigma = \frac{1}{2} \frac{n_\sigma^f - 1/2}{(1 - n_\sigma^f) n_\sigma^f}$. Finally, we introduce the Lagrangian multiplier $\lambda_\sigma$ to remove the unphysical Hilbert space (see Supplementary Note 6). The pseudo-spin carries the U(1) charge degree of freedom; the quasiparticle weight associated with the coherent part near the Fermi level is described by $Z = \langle O_\sigma \rangle \langle O_\sigma^\dagger \rangle$. An (orbital-selective) Mott localization transition happens when some quasiparticle weight $Z$ goes to zero.

In addition to the coherent quasiparticle peak, the SS method also calculates the contributions from the incoherent excitations. The Green's function of the $d$ electron $G_d$ is obtained by the convolution of the SS and the auxiliary fermions, with $G(\mathbf{k}, i\omega_n) = \sum_{i\Omega_m} G_S(i\Omega_m) G_f(\mathbf{k}, i\omega_n - i\Omega_m)$, where $G_S(\tau) = \langle T_\tau o_\sigma(\tau) o_\sigma^\dagger(0) \rangle$ and $G_f(\tau) = -\langle T_\tau f_\sigma(\tau) f_\sigma^\dagger(0) \rangle$. The spectral function is then obtained from $A(\mathbf{k}, \omega) = \frac{-1}{\pi} \Im G_d^R(\mathbf{k}, \omega)$.

## Data availability
The data that support the findings of this study are either presented in the manuscript or available at https://doi.org/10.5281/zenodo.11247849.

## Code availability
The computer codes that were used to generate the data that support the findings of this study are available from the corresponding authors upon request.

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

## Acknowledgements

We thank Gabriel Aeppli, Joseph Checkelsky, Han Wu, Yonglong Xie, and Ming Yi for useful discussions. Work at Rice has primarily been supported by the U.S. DOE, BES, under Award No. DE-SC0018197 (model construction, L.C., F.X., S.S.), by the Air Force Office of Scientific Research under Grant No. FA9550-21-1-0356 (orbital-selective Mott transition, L.C., F.X., S.S., H.H., Q.S.), by the Robert A. Welch Foundation Grant No. C-1411 (model calculation, L.C.), and by the Vannevar Bush Faculty Fellowship ONR-VB N00014-23-1-2870 (conceptualization, Q.S.).The majority of the computational calculations have been performed on the Shared University Grid at Rice funded by NSF under Grant EIA-0216467, a partnership between Rice University, Sun Microsystems, and Sigma Solutions, Inc., the Big-Data Private-Cloud Research Cyberinfrastructure MRI-award funded by NSF under Grant No. CNS-1338099, and the Extreme Science and Engineering Discovery Environment (XSEDE) by NSF under Grant No. DMR170109. H.H. acknowledges the partial support of the European Research Council (ERC) under the European Union's Horizon 2020 research and innovation program (Grant Agreement No. 101020833). Work in Vienna was supported by the Austrian Science Fund (project I 5868-N - FOR 5249 - QUAST) and the ERC (Advanced Grant CorMeTop, No. 101055088). J.C. acknowledges the support of the National Science Foundation under Grant No. DMR-1942447, support from the Alfred P. Sloan Foundation through a Sloan Research Fellowship and the support of the Flatiron Institute, a division of the Simons Foundation. J.C. and Q.S. acknowledge the hospitality of the Aspen Center for Physics, which is supported by NSF grant No. PHY-2210452.

## Author contributions

Q.S. conceived the research. L.C., F.X., S.S., H.H., J.C., and Q.S. carried out model studies. L.C., S.P., J.C., and Q.S. contributed to the development of the design principle for correlated topological phases in $d$-electron systems. L.C. and Q.S. wrote the manuscript, with inputs from all authors.

## Competing interests

The authors declare no competing interests.
