## [Peer Review File · Nature Communications]

REVIEWER COMMENTS

Reviewer #1 (Remarks to the Author):

In the manuscript, the authors have shown that one can have a nearly flat band at the Fermi level in a strongly correlated system, where Kondo-like physics is involved via the orbital-selective Coulomb interactions. To this end, they have solved the Hubbard model in the Clover lattice, which is a variant of the Kagome lattice. They have shown that in a certain range of the strength of the Coulomb interaction, a nearly flat band exists at the Fermi level. The flat band has a band crossing with another wide band, which realizes a Weyl Kondo semimetal. The authors claim that the work motivates a new material design of topological systems such as topological semimetals and fractional Chern insulators.

The manuscript suggests an interesting point that a flat band can be located at the Fermi level without being collapsed due to the strong Coulomb interaction. However, I'm reluctant to recommend the manuscript for publication in this journal for the following reasons.

1. The Clover model used in the analysis seems to have no counterpart in real materials. If the kagome lattice had been used in the study, it would have been more motivating because the kagome-like systems have been synthesized already. Moreover, since the appearance of the flat band at the Fermi level would be highly model-dependent, one cannot say that the results in the manuscript can be generally applied to other systems. In other words, the authors should suggest a possible real system that resembles the Clover lattice or propose a general principle for locating a flat band at the Fermi level in strongly correlated systems.
2. I can't entirely agree with the authors' claim that the obtained flat band could be a good platform to study many-body physics, such as the fractional topological insulators, because the Coulomb interaction is already reflected when they obtain the Kondo semimetals. Therefore, I don't think that the results in the manuscript provide a new perspective in the field of correlated topological physics.

Besides, I also have some questions as follows.

1. The flat band at $u=0$ is very similar to the flat band at $u=1.6$ in Fig. 4. Therefore, I'm a little suspicious that the flat band at $u=1.6$ is not emergent. Instead, it could be adiabatically transformed from the non-interacting flat band. The authors should clarify this issue by showing the band structures for smaller values of u so that one can trace the behavior of the non-interacting flat band as we increase u .
2. The nearly flat bands in Fig.4 seem to have a kink at the K point. Is it just a problem of the resolution of the plot?

Reviewer #2 (Remarks to the Author):

In this work, Chen et al propose a theoretical mechanism to facilitate the emergence of a flat band and a correspondingly topological Kondo semimetal, with relevant crossings appearing at the Fermi energy in d-electron systems.

The setup involves a flat band with a band width of D_{flat} , which is possibly far from the Fermi energy in the non-interacting limit, accompanied by strongly dispersing bands with a much larger band width of D_{wide} . Through the Wannier orbitals, one can define well-localized electrons that account for most of the states of the flat bands, referred to as "d" electrons, and the other Wannier orbitals mostly correspond to the states in the wide bands, referred to as "c" electrons. The authors claim that the dominant interaction effects in these degrees of freedom are the local interactions among the d electrons due to their localized nature and the flatness of the associated bands. In the regime of $D_{\text{flat}} < u < D_{\text{wide}}$, where u denotes the projected effective on-site interaction among the d electrons, an orbital selective Mott transition can be anticipated. For interaction strengths below the transition, such as $u=1.6$ in the manuscript, the coherent part of the d-electron spectral weight can be interpreted as an emergent flat band that is located around the Fermi energy. The symmetry or topologically protected crossing of this flat band with the other dispersion c-electron bands leads to topological semimetals, which can be further tuned by a Zeeman field.

If the claims made by the authors are substantiated, this paper is an interesting proposal that could spur further theoretical and experimental investigations on how the mechanism could be realized in d-electron materials. However, the current referee has a concern with the assertion that the emergent flat band is pinned to the Fermi energy.

The original energy separation between the non-interacting flat band and the Fermi energy is another energy scale that should be compared to the effective interaction strength u . From the results presented in the manuscript, one can infer that this energy scale is around 0.3, denoted as ϵ_d^0 in Fig. 1a. The smallest non-zero interaction strength of $u=1.6$ discussed in the manuscript is substantially bigger than ϵ_d^0 . From the perspective of spectral weight transfer, the favorable energetic arrangement emphasized by the authors is possible only when $u \gg \epsilon_d^0$, but at the same time, it has to be small enough to prevent crossing the orbital-selective Mott transition.

The data presented in the manuscript are insufficient to substantiate the claim that the emergent flat band is pinned to the Fermi energy even when u is continuously adjusted from small values (smaller than or comparable to ϵ_{d^0}) to $u \gg \epsilon_{d^0}$, but before the orbital-selective Mott transition. As the authors mentioned, one major motivation of the work is that the d-electron flat bands in the non-interacting limit are "generically located far away from the Fermi energy, which limits their capability to partake in the low-energy physics." It is important to clarify if the proposed mechanism really works when the non-interacting starting point indeed has flat bands that are "located far away from the Fermi energy." Otherwise, if the general results presented only hold within a relatively narrow range of projected interaction strength u , the impact of this theoretical proposal will be significantly limited.

Re: “Emergent flat band and topological Kondo semimetal driven by orbital-selective correlations”
Manuscript NCOMMS-23-08372-T
Authors: Lei Chen, Fang Xie, Shouvik Sur, Haoyu Hu, Silke Paschen, Jennifer Cano, and Qimiao Si

Preamble

We are grateful to both Reviewers who recognized the significance of our work and, in addition, to Reviewer #2 who expressed that our work “could stimulate further theoretical and experimental investigations on how the mechanism could be realized in d-electron materials,”

The Reviewers raised several concerns, all of which are very constructive and are helpful to improving our manuscript. Of these, we note in particular on two key points:

- Reviewer #1 raised the question about the generality of our results. In response, we have i) included a discussion on why we can expect our results to be valid for general frustrated-lattice systems that host flat bands (p.7, main text). Equally important, we have added the results of additional studies: Following the Reviewer’s suggestion, the added calculations (SI, the new Sec. VIII) concern the kagome lattice and show that our conclusions apply to that case as well. Both the added discussion and the new calculations reinforce our consideration and conclusion that the results are generally valid to geometry-induced flat band systems.
- Reviewer #2 noted on the role of energy and interaction scales. In response, here too we have added the results of new calculations (SI, the new Sec. VII). We constructed a phase diagram by scanning the parameters ϵ_d^0 and u , and explicitly showed that there exists a large parameter region for the physics we claimed; this parameter regime corresponds to where the correlation effects are strongly orbital-selective, as the title of our manuscript implies.

We would like to add that the theoretical notion we have introduced in this work has already motivated new experiments. In a work that involves several of us, a new experiment shows in a frustrated-lattice material that an emergent flat band develops at the Fermi energy even though the *ab initio* noninteracting bandstructure predicts a flat band that is considerably away from the Fermi energy. We have now included a “Note added” in the manuscript on this experimental evidence for our theory.

We now turn to a point-by-point response along with a summary of the main changes. With these improvements, we believe our manuscript is now suited for publication in Nature Communications.

Reply to Reviewer #1

REVIEWER: *In the manuscript, the authors have shown that one can have a nearly flat band at the Fermi level in a strongly correlated system, where Kondo-like physics is involved via the orbital-selective Coulomb interactions. To this end, they have solved the Hubbard model in the Clover lattice, which is a variant of the Kagome lattice. They have shown that in a certain range of the strength of the Coulomb interaction, a nearly flat band exists at the Fermi level. The flat band has a band crossing with another wide band, which realizes a Weyl Kondo semimetal. The authors claim that the work motivates a new material design of topological systems such as topological semimetals and fractional Chern insulators.*

The manuscript suggests an interesting point that a flat band can be located at the Fermi level without being collapsed due to the strong Coulomb interaction. However, I’m reluctant to recommend the manuscript for publication in this journal for the following reasons.

Reply: We would like to thank the Reviewer for recognizing the significance of our work and for a concise summary of our research. In addressing the concerns raised by the Reviewer, please refer to our responses below point-by-point.

REVIEWER: 1. The Clover model used in the analysis seems to have no counterpart in real materials. If the kagome lattice had been used in the study, it would have been more motivating because the kagome-like systems have been synthesized already. Moreover, since the appearance of the flat band at the Fermi level would be highly model-dependent, one cannot say that the results in the manuscript can be generally applied to other systems. In other words, the authors should suggest a possible real system that resembles the Clover lattice or propose a general principle for locating a flat band at the Fermi level in strongly correlated systems.

Reply: We thank the Reviewer for this very valuable comment and suggestion. We will address it as follows.

Firstly, as a matter of fact, the lattice with a clover-lattice-like structure occurs in real materials, as seen in H. Wu et al. (the new Ref. [30] of the manuscript).

Secondly, we appreciate the Reviewer's suggestion about the case of a kagome lattice, and have now added calculations for the emergent flat bands in this case (Fig.R1 and the SI, Fig.S7). We find that similar results regarding the pinning of the flat band at the Fermi energy holds in for the kagome case.

Figure R1: **Topological Kondo semimetal in a kagome lattice.** **a**, The noninteracting band structure. **b**, The dispersion of the coherent single-electron excitations at $u = 2.1$. The red solid curve denotes the emergent flat band close to the Fermi energy. The grey lines mark the incoherent single-electron excitations. **c**, The zoomed-in view of the emergent flat band. **d**, The band structure at $u = 2.1$ with a Zeeman splitting $m_z = 0.05$.

Here, we consider a two-orbital model defined on the kagome lattice and extract the flat band from the middle of the whole band structure, along with the nearby wide band which is right on the top of the flat band for Wannier construction (Ref. [57]). As shown in Fig. R1 (a), we consider the model that, in the noninteracting limit, the flat band is considerably (about 0.45) away from the Fermi energy. The electron correlation causes an emergent flat band that is pinned to near the Fermi energy as illustrated in Fig. R1

(b). With the spin-orbit coupling included in this kagome model, the Dirac node between the flat band and the dispersive band is gapped out, but a relatively small Zeeman coupling leads to a substantial change in the topological properties of the bands. As depicted in Fig. R1 (d), with a small Zeeman coupling, the topological gap closes and nodal lines, which encircle the Γ point, appear. They are protected by the mirror M_z symmetry.

Thirdly, this mechanism can also be viewed from a Kondo perspective as appropriate for the regime with strong orbital selectivity. The emergent flat band can be viewed as describing low energy electronic excitations associated with the Kondo-driven composite fermions. Because low-energy coherent electronic excitations are always Fermi-energy bound, and also based on well-established understanding that Kondo-driven composite fermions occur in the immediate vicinity of the Fermi energy, the emergent flat bands that develop through our proposed mechanism must be pinned near the Fermi energy. This represents “a general principle”, at the level that the Reviewer asked for, which is supported by the explicit calculations we have reported for both the clover and kagome lattices.

Finally, as a general principle, we expect that our results will apply to other frustrated lattices that promote flat bands, such as pyrochlore lattice, although explicit calculations in that case are more difficult. Indeed, as alluded to in the preamble, our theory has already successfully motivated a new experiment that observed an emergent flat band precisely at the Fermi energy even though the DFT flat band is about 0.5 eV away from the Fermi energy.

Changes to the manuscript: To explicitly show the generality of our observation, we’ve added a new section (Sec. VIII) in the SI to report the results on the orbital selective correlation that happens in a kagome flat band system. In addition, we have added the general argument on the robustness of our conclusion in the main text (p.6), where we also mention the new kagome result (p.7).

REVIEWER: *2. I can’t entirely agree with the authors’ claim that the obtained flat band could be a good platform to study many-body physics, such as the fractional topological insulators, because the Coulomb interaction is already reflected when they obtain the Kondo semimetals. Therefore, I don’t think that the results in the manuscript provide a new perspective in the field of correlated topological physics.*

Reply: We thank the reviewer for raising this important point. A necessary condition for realizing a fractional Chern insulator or other topological systems with fractional charge is that the flat band needs to cross the Fermi energy. Our theory aims to address this necessary condition first. Regarding the concern that “the Coulomb interaction is already reflected when they obtain the Kondo semimetals,” in the current calculation, we are using an analysis of the interaction effects at the saddle-point level; this analysis captures the Kondo effect, which gives rise to the Kondo-driven flat band pinned near the Fermi energy. Beyond the saddle-point level, there are residual interaction effects. The analysis of such residual interactions is required to establish a concrete connection between the current scheme and a fractional Chern insulator. While this is beyond the scope of our current study, we do feel that it is physically plausible and it is useful to discuss this effect in our manuscript in light of the topological nature of the emergent flat band.

Changes to the manuscript: Where the manuscript refers to the residual interactions (p.8), we have now specified that such interactions “develop beyond the saddle-point analysis we have carried out”. In the same paragraph (p.7), we also added “1/3-filling” as an illustration for the filling factor when previously this was only referred to very generally.

REVIEWER: *Besides, I also have some questions as follows. 1. The flat band at $u=0$ is very similar to the flat band at $u=1.6$ in Fig. 4. Therefore, I’m a little suspicious that the flat band at $u=1.6$ is not emergent. Instead, it could be adiabatically transformed from the non-interacting flat band. The authors should clarify this issue by showing the band structures for smaller values of u so that one can trace the*

behavior of the non-interacting flat band as we increase u .

Reply: The typical way to discuss a Kondo system begins with the strong interaction limit where the more strongly correlated electrons are considered to be a local moment. We then consider the problem of the local spins coupled to the less correlated electrons, which serve as the conduction electrons. The Kondo effect leads to the development of a composite fermion, which dominates the lower energy physics. The band, developed by the composite fermion, is the emergent flat band of our focus. It appears when the system's correlations are sufficiently strong such that it experiences strong orbital selectivity, but are not too strong so that it has not yet reached the orbital selective Mott phase (“Kondo-destroyed” phase). In other words, the correlations place the system on the itinerant side, but in *proximity* to the orbital selective Mott transition point. On the side of “Kondo screened” phase, with decreasing interactions, there is a crossover between regimes though no thermodynamic phase transition.

Figure R2: (a) The quasiparticle weight of the more strongly correlated species of electrons (d) versus the interaction strength and for various ϵ_d^0 and (b) the corresponding colormap plot. In the parameter setting between the two black lines (region “II”), an emergent flat band develops in the immediate vicinity of the Fermi energy. (c)-(e) The dispersion of the coherent single-electron excitations are shown for parameters corresponding to the purple points marked in (b); (f)-(h) the counterparts for the red points marked in (b).

Therefore, an instructive way to consider the word “emergent” is to compare the dispersion of the low-energy coherent single-electron excitations when u is sufficiently large: For the d -electron systems we consider, the analogue of the Kondo regime of the f -electron systems is the regime of a correlated metal in which the correlation effects are strongly orbital-selective; this is the regime where a flat band emerges at the Fermi energy. The title of our manuscript already specifies this.

To make these considerations concrete, we have now added calculations by scanning the phase diagram as a function of ϵ_d^0 and u . The quasiparticle weight of the more strongly correlated species of the electron is shown in Fig. R2 (a) accompanied with the corresponding colormap in Fig. R2 (b). The region where the Kondo physics – captured by the strong orbital selectivity– develops is between the two black dash lines (region “II”), when the quasiparticle weight is sufficiently decreased from 1. If the interaction u is below the lower dash line, there is no flat band near the Fermi energy, and if the interaction u is larger than the higher dash line, the more strongly correlated species of electrons become fully localized. This can be further seen by comparing the single particle excitations from Fig. R2 (c-h), which correspond to the parameter settings marked in purple and red in Fig. R2 (b).

In short, the regime with an emergent flat band applies to the entire green region (i.e., region “II”) between the two dashed lines of Fig. R2 (b), while the non-interacting limit belongs to the yellow regime (i.e., region “III”) of Fig. R2 (a). It is only in the green region that the notion of emergent flat band applies.

Changes to the manuscript: Following also the comments of Reviewer #2, we added the details of the aforementioned new calculations as a new section (Sec. VII) in the SI, which demonstrates that an emergent flat band develops over an extended parameter regime (region “II” of the overall phase diagram) and that this regime is distinct from the noninteracting limit (which appears in region “III” of the overall phase diagram). This also is now briefly discussed in the main text (p.7).

2. *The nearly flat bands in Fig.4 seem to have a kink at the K point. Is it just a problem of the resolution of the plot?*

Reply: The kink appears because we introduced a term to break the C_3 symmetry, resulting in different velocities along the $\Gamma - K$ and $K - M$ directions. This feature is already present in the non-interacting band structure. A zoomed-in figure for original band structure is shown in Fig. R3.

Figure R3: Zoomed-in dispersion for the original band structure.

Reply to Reviewer #2

REVIEWER: *In this work, Chen et al propose a theoretical mechanism to facilitate the emergence of a flat band and a correspondingly topological Kondo semimetal, with relevant crossings appearing at the Fermi energy in d-electron systems.*

The setup involves a flat band with a band width of D_{flat} , which is possibly far from the Fermi energy in the non-interacting limit, accompanied by strongly dispersing bands with a much larger band width of D_{wide} . Through the Wannier orbitals, one can define well-localized electrons that account for most of the states of the flat bands, referred to as “d” electrons, and the other Wannier orbitals mostly correspond to the states in the wide bands, referred to as “c” electrons. The authors claim that the dominant interaction effects in these degrees of freedom are the local interactions among the d electrons due to their localized nature and the flatness of the associated bands. In the regime of $D_{\text{flat}} < u < D_{\text{wide}}$, where u denotes the projected effective on-site interaction among the d electrons, an orbital selective Mott transition can be anticipated. For interaction strengths below the transition, such as $u=1.6$ in the manuscript, the coherent part of the d-electron spectral weight can be interpreted as an emergent flat band that is located around the Fermi energy. The symmetry or topologically protected crossing of this flat band with the other dispersion c-electron bands leads to topological semimetals, which can be further tuned by a Zeeman field.

Reply: We thank the Reviewer for a comprehensive summary of the key elements of our paper.

REVIEWER:*If the claims made by the authors are substantiated, this paper is an interesting proposal that could spur further theoretical and experimental investigations on how the mechanism could be realized in d-electron materials. However, the current referee has a concern with the assertion that the emergent flat band is pinned to the Fermi energy.*

Reply: We thank the Reviewer for recognizing the significance and potential impact of our work. Please see the following for a detailed explanation for the expressed concern.

REVIEWER: *The original energy separation between the non-interacting flat band and the Fermi energy is another energy scale that should be compared to the effective interaction strength u . From the results presented in the manuscript, one can infer that this energy scale is around 0.3, denoted as ϵ_d^0 in Fig. 1a. The smallest non-zero interaction strength of $u=1.6$ discussed in the manuscript is substantially bigger than ϵ_d^0 . From the perspective of spectral weight transfer, the favorable energetic arrangement emphasized by the authors is possible only when $u \gg \epsilon_d^0$, but at the same time, it has to be small enough to prevent crossing the orbital-selective Mott transition.*

The data presented in the manuscript are insufficient to substantiate the claim that the emergent flat band is pinned to the Fermi energy even when u is continuously adjusted from small values (smaller than or comparable to ϵ_d^0) to $u \gg \epsilon_d^0$, but before the orbital-selective Mott transition. As the authors mentioned, one major motivation of the work is that the d-electron flat bands in the non-interacting limit are “generically located far away from the Fermi energy, which limits their capability to partake in the low-energy physics.” It is important to clarify if the proposed mechanism really works when the non-interacting starting point indeed has flat bands that are “located far away from the Fermi energy.” Otherwise, if the general results presented only hold within a relatively narrow range of projected interaction strength u , the impact of this theoretical proposal will be significantly limited.

Reply: We thank the reviewer for pointing out a very relevant energy scale, which is the energy level (ϵ_d^0) of the flat band. From the perspective of Kondo physics, ϵ_d^0 exactly corresponds to the local energy of the f electron. In a typical heavy fermion system, considering an Anderson lattice model, if we set the local energy $\epsilon_d^0 = -U/2$ and both are large compared to the f -electron bandwidth and the hybridization, the Kondo effect develops. When the condition $\epsilon_d^0 = -U/2$ is not satisfied, there is a correspondingly extended regime of the parameter space for Kondo effect under a particle-hole asymmetric setting; the extended na-

ture of this parameter space is enabled by the narrowness of the f -electron dispersion.

The counterpart of the above for the flat band system is that, so long as the flat band width D_{flat} is small, the orbital-selective correlation regime occurs over an extended parameter space and this makes the emergent flat band notion robust. The added calculation, described in Fig. R2 and in the SI, Fig. S6, illustrate this point: The regime where the emergent flat band occurs corresponds to a wide parameter regime marked by region “II” of Fig. R2. Thus, the notion does not require fine-tuning of parameters.

Changes to the manuscript: A full new section (Sec. VII) has been added in the SI to further elaborate on the overall phase diagram in the parameter space for the local energy level and interaction strength. That an emergent flat band develops over a large parameter space in the overall phase diagram is now also discussed in the main text (pp. 6-7).

REVIEWER COMMENTS

Reviewer #1 (Remarks to the Author):

In the revised manuscript, the authors have diligently addressed most of the referee's concerns through an extensive additional analysis. Notably, in response to the first question regarding the generalization of the results on the Clover lattice, the authors have undertaken a comprehensive examination of a distinct model – the Kagome lattice model. Their thoughtful exploration and persuasive arguments regarding the broader applicability of their original study on the Clover lattice have left a positive impression on me. Consequently, I am inclined to support the publication of the revised manuscript in this journal.

However, before finalizing my recommendation, I have one specific request. The aspect concerning "residual interactions" is important, given the author's assertion that the "Kondo-driven flat band" constitutes a novel platform for many-body studies, a crucial conclusion of the manuscript. While I understand the authors' claim that this topic might be considered beyond the paper's scope, I believe a more detailed exposition on "residual interactions" is necessary.

I propose that, instead of giving a full solution, the authors provide a glimpse of the expected form of these "residual interactions." This addition, even in a simplified manner, would enhance the manuscript by addressing the crucial aspect of the Kondo-driven flat band and its implications. If this request is properly answered, I would be more than inclined to recommend the manuscript for publication in this journal.

Reviewer #2 (Remarks to the Author):

I have reviewed the previous report by the other referee and the authors' responses to the reports. I am satisfied with the authors' revisions and response, and I am convinced that the reported emergence of correlated flat bands is a general phenomenon in a broad range of systems with various underlying lattices. This finding is a valuable contribution to the literature, especially given the current significant efforts to realize correlated flat band materials. Therefore, I recommend that the work be published in Nature Communications.

Re: “Emergent flat band and topological Kondo semimetal driven by orbital-selective correlations”
Manuscript NCOMMS-23-08372-A
Authors: Lei Chen, Fang Xie, Shouvik Sur, Haoyu Hu, Silke Paschen, Jennifer Cano, and Qimiao Si

Preamble

We are grateful to both Reviewers for their thoughtful comments about our manuscript. Reviewer #2 has recommended it for publication. Reviewer #1 is likewise enthusiastic about it, and furthermore made a constructive suggestion that we give “a glimpse of the expected form” of the “residual interactions”. As described below, we have followed this suggestion and have made the corresponding revision to the manuscript.

With this improvement, we believe that our manuscript is now suitable for publication in Nature Communications.

Reply to Reviewer #1

REVIEWER: *The aspect concerning “residual interactions” is important, given the author’s assertion that the “Kondo-driven flat band” constitutes a novel platform for many-body studies, a crucial conclusion of the manuscript. While I understand the authors’ claim that this topic might be considered beyond the paper’s scope, I believe a more detailed exposition on “residual interactions” is necessary.*

I propose that, instead of giving a full solution, the authors provide a glimpse of the expected form of these “residual interactions.” This addition, even in a simplified manner, would enhance the manuscript by addressing the crucial aspect of the Kondo-driven flat band and its implications. If this request is properly answered, I would be more than inclined to recommend the manuscript for publication in this journal.

Reply: We thank the reviewer for raising this important point. We have so far considered the dominant terms of the interactions that are projected into the Wannier basis: the on-site Coulomb interaction among the most localized molecular (Wannier) orbitals (d) and the other on-site interactions between the d orbitals and those of the more extended molecular orbitals (c_1 and c_2), as described in Eqs. (1)(6). Taking into account these interactions at the saddle-point level in the slave-spin approach leads to the emergent flat band, in which the (renormalized) electronic quasiparticle states are primarily associated with the effective d orbitals. The residual interactions for the emergent flat bands are mainly of three types, all of which are primarily pertinent to the d electrons (as opposed to the c_1 and c_2 electrons):

The first type is the Ruderman–Kittel–Kasuya–Yosida (RKKY) interaction between the d electrons, mediated by the conduction c electrons. For the generic case with typical fillings of the conduction c electrons, the RKKY interaction tends to be antiferromagnetic. However, when the c electrons are dilute, the RKKY interaction would be ferromagnetic.

The second type is the superexchange interaction between the d electrons. This arises because, in the effective (projected) model, the off-site hybridization between the d orbitals is non-zero and so is the hybridization between the d and c orbitals. These kinetic terms induce a coupling between the low-energy renormalized (i.e., coherent) part of the d -electron spectral weight near the Fermi energy and the high-energy incoherent part of the d -electron spectral weight (which is controlled by both u -interaction and bare d -level); when the latter are integrated out, the superexchange interaction ensues. The superexchange interaction is typically antiferromagnetic and is maximized near the (orbital-selective) Mott transition point. Deep in a Mott insulating phase, the value of the superexchange interaction can be derived using a direct perturbation theory in terms of the d - d hopping and d - c hybridization. In the bad metal region, i.e. in the regime of our interest—where the coherent part of the d -electron spectral weight is nonzero but small, such that the system is in proximity to the orbital-selective Mott transition – the superexchange interaction can be constructed by considering the fluctuations beyond the saddle-point level such that the incoherent part of the slave-spin spectral weight is taken into account: This method is in parallel to the derivation for the superexchange

interaction in a closely-related slave-particle representation, viz. the slave-rotor approach [See Fig. 2(b) of Phys. Rev. B 100, 235113 (2019), which is co-authored by one of the current authors].

The third type of residual interactions consists of the direct exchange among the d -electrons. They can be constructed from the projection of the original Hubbard interaction to the Wannier basis. Although the emergent molecular orbitals are exponentially localized, even the most compact (d) orbital still extends over several lattice sites. The shared sites between the nearby molecular orbitals contribute to this direct exchange interaction. This typically ferromagnetic interactions between the neighboring sites have the form of the Hund's-like two-body interactions.

Note that the processes leading to the first and third type of residual interactions naturally lead to density-density interactions between the neighboring sites as well.

In summary, the forms of the residual interactions are of the spin-spin and density-density interactions among neighboring sites. An appropriate study of the dominant residual interactions should include the above terms, and the competition between them could lead to different phases. For example, if the antiferromagnetic interactions prevail over the ferromagnetic ones, they promote antiferromagnetic-ordering tendencies and the associated quantum criticality. If the ferromagnetic interactions dominate, they may split the Kramers degeneracy, leading to flat bands with a non-zero Chern number which can be analyzed for, e.g., a lattice realization of fractional quantum Hall effect [Phys. Rev. Lett. 127, 246403 (2021), which is co-authored by one of the current authors]. Indeed, as we have already noted in the previous version of the manuscript, the combination of the flatness of the associated bands and the aforementioned residual interactions represents a condition that is similar to what happens in the moiré systems, though the Z_2 nature of the flat bands makes our case distinct and rare. A detailed analysis of the effect of the residual interactions is beyond the scope of the present work.

Changes to the manuscript: A new section (Sec. V) has been added in the SI to describe the form of the residual interactions and the possible consequences resulting from the competition between them. The discussion about the FCI states in the main text (p.8) is appropriately modified, and several supporting references (Refs. 44 and 55) are added.